# BrtB is an O-alkylating enzyme that generates fatty acid-bartoloside esters

João P.A. Reis [1], Sandra A.C. Figueiredo[1], Maria Lígia Sousa [1] & Pedro N. Leão [1✉]

Esterification reactions are central to many aspects of industrial and biological chemistry. The formation of carboxyesters typically occurs through nucleophilic attack of an alcohol onto the carboxylate carbon. Under certain conditions employed in organic synthesis, the carboxylate nucleophile can be alkylated to generate esters from alkyl halides, but this reaction has only been observed transiently in enzymatic chemistry. Here, we report a carboxylate alkylating enzyme – BrtB – that catalyzes O-C bond formation between free fatty acids of varying chain length and the secondary alkyl halide moieties found in the bartolosides. Guided by this reactivity, we uncovered a variety of natural fatty acid-bartoloside esters, previously unrecognized products of the bartoloside biosynthetic gene cluster.

---

[1] Interdisciplinary Centre of Marine and Environmental Research (CIIMAR/CIMAR), University of Porto, Avenida General Norton de Matos, s/n, 4450-208, Matosinhos, Portugal. ✉email: pleao@ciimar.up.pt

sters, in particular carboxyesters, are fundamental to both industrial[1] and biological[2] chemistry. A number of synthetic methods have been developed to generate carboxyesters, with most exploiting nucleophilic attack of an alcohol onto the carboxylic carbon[3], as typified by the century-old Fischer-Speier esterification[4]. Esterification reactions that are independent of alcohols and in which the carboxylate anion acts as nucleophile are also well-known in organic synthesis, in particular with primary alkyl halides as the electrophilic partner[3]. Likewise, biological carboxyester formation most often takes advantage of the electrophilicity of a carbonyl group, via acylation of a suitable alcohol (e.g., lipases and esterases[2], thioesterases[5], condensation of α-hydroxy acids in non-ribosomal peptide synthetases[6]), or through direct oxidation by Baeyer-Villiger monooxygenases[7]. Carboxylate nucleophiles can also give rise to carboxyesters through SAM-dependent methylation of carboxylates, e.g., ref. [8] or intramolecular esterifications as those catalyzed by the 3-carboxy-*cis,cis*-muconate lactonizing enzyme (CMLE)[9] or the more recently described radical S-adenosylmethionine (rSAM) NosN[10] (Fig. 1a–c). However, to our knowledge, enzymatic O–C bond formation involving a carboxylate nucleophile and an alkyl halide electrophile has only been reported to occur transiently during catalysis by haloalkane dehalogenases and certain haloacid dehalogenases[11]. In their catalytic cycles, an ester bond is formed through attack of a side-chain carboxylate onto a halogenated position and is subsequently hydrolyzed to generate an alcohol product (Fig. 1d).

Recently, Balskus and co-workers unveiled a biological C-alkylation involved in the biosynthesis of cyclophane natural products in cyanobacteria[12]. This reaction requires the previous stereoselective chlorination of an unactivated carbon center by the CylC halogenase. To create the final cyclophane dimeric scaffold, an alkylating enzyme (CylK) catalyzes C–C bond formation between C-2 of each alkylresorcinol monomer and the halogenated carbon of the other monomer (Fig. 1e)[12]. A number of cyanobacterial biosynthetic gene clusters (BGCs) were found to feature both CylC and CylK homologs, suggesting that analogous C–C bond formation could be a common feature of secondary metabolite biosynthesis in cyanobacteria[12]. One such BGC—*brt* —encodes the biosynthesis of the bartolosides, a group of chlorinated dialkylresorcinols[13,14]. In *brt* clusters, the homolog of the CylC halogenase (BrtJ) is likely responsible for the mid-chain chlorinations that are present in all bartolosides (Fig. 1f)[12–14]. However, the role played by the *brt*-encoded CylK homolog (BrtB) has remained unclear. The reasons for this are twofold: first, the action of a CylK homolog is not necessary to explain the biosynthesis of currently known bartolosides[13,14]; second, CylK requires a free C-2 resorcinol as a nucleophile[12,15], a position that is unavailable (it is alkylated) in the bartolosides. Here, we show that BrtB is an O-alkylating enzyme that catalyzes the esterification of free (non-activated) fatty acids with the chlorinated positions found in the bartolosides, generating a diversity of fatty acid-bartoloside esters (Fig. 1g).

## Results

### Bartoloside esters are formed upon fatty acid supplementation.

The biosynthesis of the dialkylresorcinol skeleton in the bartolosides involves recruitment of fatty acid derivatives from primary metabolism[13]. We envisioned that this could be exploited to incorporate terminal alkyne moieties into the bartolosides and generate click chemistry-accessible versions of these natural products for probing their biological role, e.g., ref. [16]. To this end, we supplemented cultures of the bartoloside A (**1**)-producing cyanobacterium *Synechocystis salina* LEGE 06099 with 50 mg L$^{-1}$ of 5-hexynoic or 6-heptynoic (**2**) acids. LC-HRESIMS analysis of

the resulting cell extracts revealed a massive depletion of the major metabolite **1** and several of its analogues (Fig. 2a, Supplementary Fig. 1) in supplemented cultures, yet, surprisingly, we did not detect any ions compatible with bartolosides containing terminal alkynes in their dialkylresorcinol skeleton. Instead, we observed a series of *m/z* values consistent with the incorporation of one or two units of the supplemented fatty acids into the depleted bartolosides and the concomitant loss of one or both Cl atoms, respectively (Fig. 2a, Supplementary Fig. 2). This was supported by LC-HRESIMS/MS analysis of these species, which showed fragments corresponding to the intact alkyne precursors or to their neutral losses (Supplementary Fig. 2). To unequivocally establish the identity of the newly observed compounds, we isolated two major metabolites (**3** and **4**) resulting from the supplementation with **2**. Subsequent structure elucidation using 1D and 2D NMR as well as HRESIMS/MS analyses, clarified that **3** and **4** were esters of bartolosides A (**1**) and G (**5**), respectively, in which **2** was esterified to the previously chlorinated positions (Supplementary Note 1, Supplementary Fig. 3). Supplementation of *S. salina* LEGE 06099 with butyric, caprylic, lauric and palmitic acids, as well as with 7-bromoheptanoic acid led to the formation of the corresponding monoesters and diesters (Supplementary Fig. 4). Overall, our results show that exogenously provided fatty acids are converted in vivo into fatty acid-bartoloside esters by *S. salina* LEGE 06099.

### BrtB esterifies free fatty acids with bartoloside A.

We set out to investigate whether BrtB, the only enzyme in the *brt* gene cluster with no ascribed function, could be responsible for ester formation. Following the protocol reported by Schultz et al.[15], we expressed and purified a Strep-Tag©-recombinant version of BrtB (NStrep-BrtB) in *Escherichia coli* BL21 DE3 Rosetta cells and tested its ability to convert **1** and **2** into **3** in vitro. We found that adding NStrep-BrtB to a reaction mixture composed of **1**, **2** and Ca$^{2+}$ and Mg$^{2+}$-containing buffer[12,15] was necessary and sufficient to generate diester **3** as well as monoester(s) **6a** and/or **6b** (Fig. 2b, Supplementary Fig. 5). In line with the in vivo data, we found that BrtB is also able to esterify palmitic acid with **1** to generate bartoloside A palmitate(s) **7a** and/or **7b** as well as bartoloside A dipalmitate (**8**) (Fig. 2c, Supplementary Fig. 6). Thus, we show that BrtB catalyzes C–O bond formation through esterification of a free fatty acid with a secondary alkyl halide.

### BrtB is a promiscuous O-alkylating enzyme.

Because our in vivo supplementation experiments showed that *S. salina* LEGE 06099 cells generated bartoloside esters of fatty acids ranging from C$_4$ to C$_{16}$ (Supplementary Fig. 4), we next tested the substrate scope of BrtB in enzymatic assays. BrtB could generate the predicted mono- and diester products for all the tested fatty acids, ranging in length from C$_2$ to C$_{16}$ (Fig. 3a, Supplementary Fig. 7) but not for other carboxylic acids (Supplementary Fig. 7). Potential nucleophiles with amide, alcohol or phosphonic acid functionalities did not lead to any observable product (Supplementary Fig. 7) when incubated with BrtB and **1**. Primary alkyl halides and an aliphatic secondary alkyl halide were not esterified with **2** either (Supplementary Fig. 7). We additionally tested BrtB activity at 15, 25, 37, and 55 °C and observed formation of substrate in all tested temperatures, with maximum activity at 37 °C (Supplementary Fig. 8). Having found that BrtB could use acetate as a substrate, we performed in vitro assays with [$^{18}O_2$]acetic acid to gain some mechanistic insight into the reaction. As expected, LC-HRESIMS/MS analysis revealed that two or four $^{18}O$ atoms were incorporated into the corresponding bartoloside mono- and diesters, respectively (Fig. 3b). The fragmentation pattern

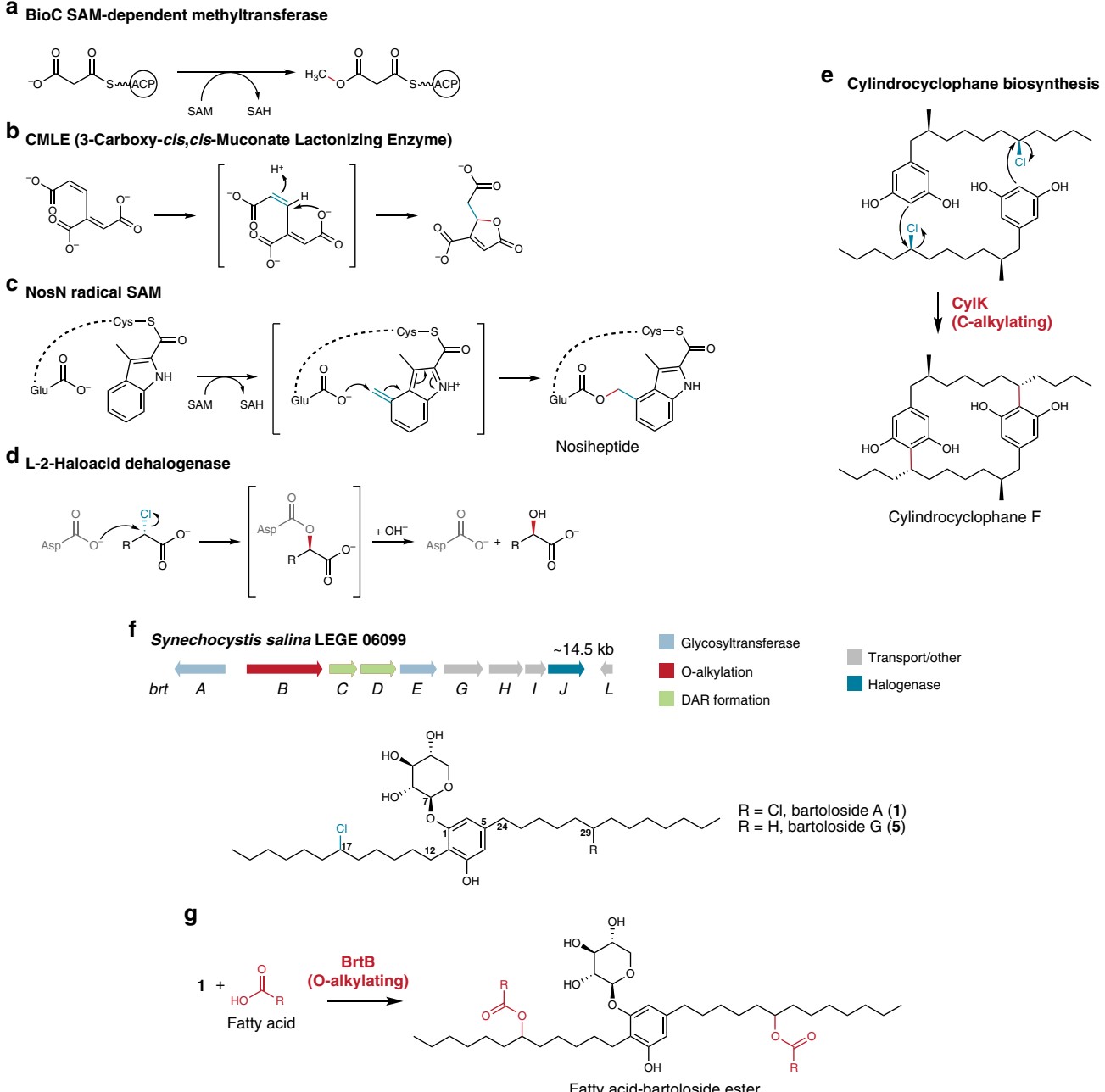

**Fig. 1 Selected O- and C-alkylating enzymes. a** BioC is a SAM-dependent methyltransferase acting in biotin biosynthesis. **b** CMLE is an eukaryotic beta-propeller enzyme catalyzing an intramolecular addition of a carboxylate to a double bond. **c** NosN, a radical SAM enzyme, generates an exomethylene intermediate that drives intramolecular ester bond formation. **d** Dehalogenation of alkyl halides catalyzed by L-2-haloacid dehalogenases involves the transient formation of an ester species resulting from the nucleophilic attack of a side chain carboxylate onto the chlorinated carbon. **e** In cylindrocyclophane biosynthesis, each alkylresorcinol moiety is alkylated at the C-2 position by CylK. **f** The *brt* gene cluster from *Synechocystis salina* LEGE 06099 encodes the production of a number of bartolosides, the most abundant of which are bartolosides **1** and **5** (depicted). **g** BrtB catalyzes O–C bond formation between non-activated fatty acids and bartolosides.

indicated that these atoms were part of the esterified acetate moiety (Supplementary Fig. 9), proving that the fatty carboxylate acts as a nucleophile in this reaction. Carrying out the esterification in D$_2$O-resuspended reaction buffer did not lead to deuterium incorporation into the reaction products (Supplementary Fig. 10), dismissing the possibility of a two-step elimination and nucleophilic addition mechanism. Overall, we show that BrtB is highly promiscuous as to fatty acid length and catalyzes O–C bond formation between a fatty carboxylate nucleophile and the chlorinated carbons in the bartolosides.

**The *brt* cluster generates diverse bartoloside esters**. Having established that the observed esterification was catalyzed by a *brt*-encoded enzyme, we hypothesized that fatty acid-bartoloside esters are products of the *brt* cluster, in which case their production by cyanobacterial cells should occur without exogenous fatty acid supply. In fact, the LC-HRESIMS data for non-supplemented controls in our in vivo fatty acid supplementation experiments showed an abundant compound with *m/z* values and retention time consistent with a bartoloside A monopalmitate (**7a**/**7b**, Supplementary Fig. 4). We thoroughly explored LC-HRESIMS data

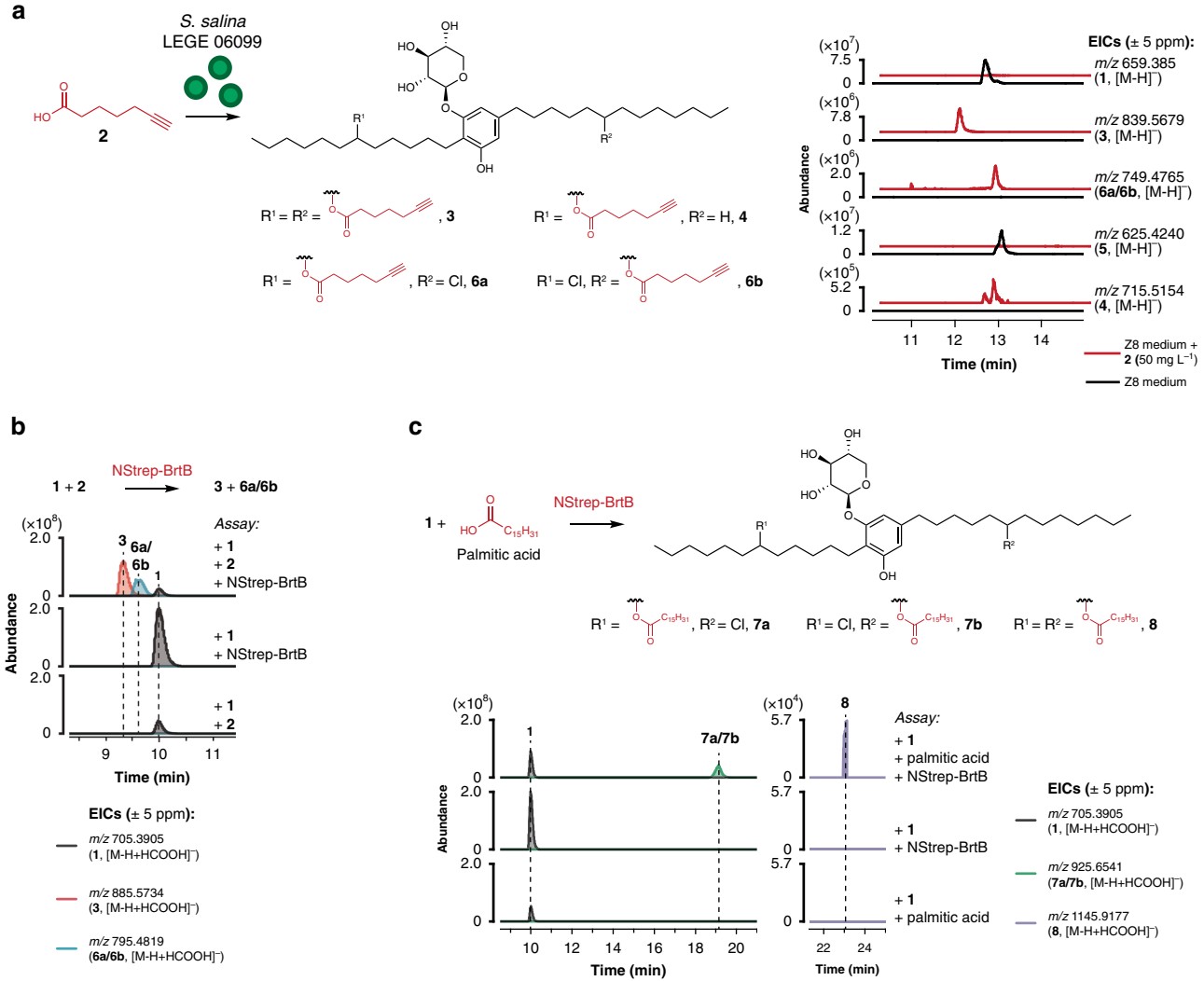

**Fig. 2 Bartoloside-fatty acid esters are formed in vivo and in vitro by BrtB.** In cultures of *S. salina* LEGE 06099 supplemented with 50 mg L$^{-1}$ of 6-heptynoic acid (**2**), formation of the bartoloside esters **3**, **6a**/**6b** and **4** with concomitant depletion of the corresponding bartolosides (A, **1** or G, **5**) was observed by LC-HRESIMS (**a**). Analysis (LC-HRESIMS) of the NStrep-BrtB-mediated O-alkylation of bartoloside A (**1**) with **2** to generate diester **3** and monoesters **6a**/**6b** (**b**) or with palmitic acid to generate monoesters **7a**/**7b** and diester **8**, the latter in much lower amount (**c**). Full reactions contained 1 μM recombinant NStrep-BrtB, 100 μM of **1** and 200 μM of either **2** (**b**) or palmitic acid (**c**).

from a CH$_2$Cl$_2$/MeOH (2:1) cellular extract of a batch culture of *S. salina* LEGE 06099 and detected species with *m/z* values and fragmentation patterns corresponding to mono- and diesters of bartolosides, the most abundant of which matched the predicted *m/z* of bartoloside A monopalmitate(s) **7a** and/or **7b**, while a peak consistent with bartoloside A dipalmitate (**8**) was also detected but less abundant (Fig. 4a). We were able to separate and isolate the major bartoloside ester **7a** as well as the less abundant **7b** and confirmed their structures through HRESIMS/MS and NMR analyses (Fig. 4b, Supplementary Note 1, Supplementary Fig. 11). To clarify the identity of additional natural bartoloside esters, we searched the LC-HRESIMS data for *m/z* values matching esters of bartolosides **1** and **5** with fatty acids known to occur in cyano-bacterial cells[17]. This revealed 18 additional bartoloside esters, of varying chain length and saturation, whose structures could be confirmed by LC-HRESIMS/MS analysis (Fig. 4c, Supplementary Figs. 12 and 13). In all probability, additional esters are produced through the action of BrtB on the minor bartolosides E, F and H-K produced by *S. salina* LEGE 06099[14], but their detection and structure determination is hampered by low levels and similar *m/z* values or retention times to those of more abundant bartoloside

esters. Neither **7a**, nor the alkyne-containing **3** and **4** showed appreciable cytotoxicity (up to 10 μM), against immortalized cell lines or antibacterial activity at 0.5 mg mL$^{-1}$ in agar disk diffusion assays (Supplementary Figs. 14 and 15). Our findings revealed that the *brt* gene cluster generates a large diversity of natural products, not only through the incorporation of fatty acid derivatives of different chain length during dialkylresorcinol formation and different halogenation patterns[13,14], but also through the relaxed specificity of BrtB, which is able to alkylate a variety of endogenous fatty acids.

**Regioselectivity and kinetics of BrtB.** Because palmitic acid and **1** were found to be abundant natural substrates of BrtB, we performed 10 h kinetic assays to determine how esterification proceeds, in particular whether BrtB had selectivity for a particular alkyl chain in the first esterification or for a particular monoester in the second esterification. LC-HRESIMS analysis of assays quenched at different time points indicated that BrtB is slightly regioselective (two-fold factor) for the C-2 alkyl chain over the C-5 alkyl chain, but performed the second esterification at similar rates with either **7a** or **7b** as substrates (Fig. 4d, e,

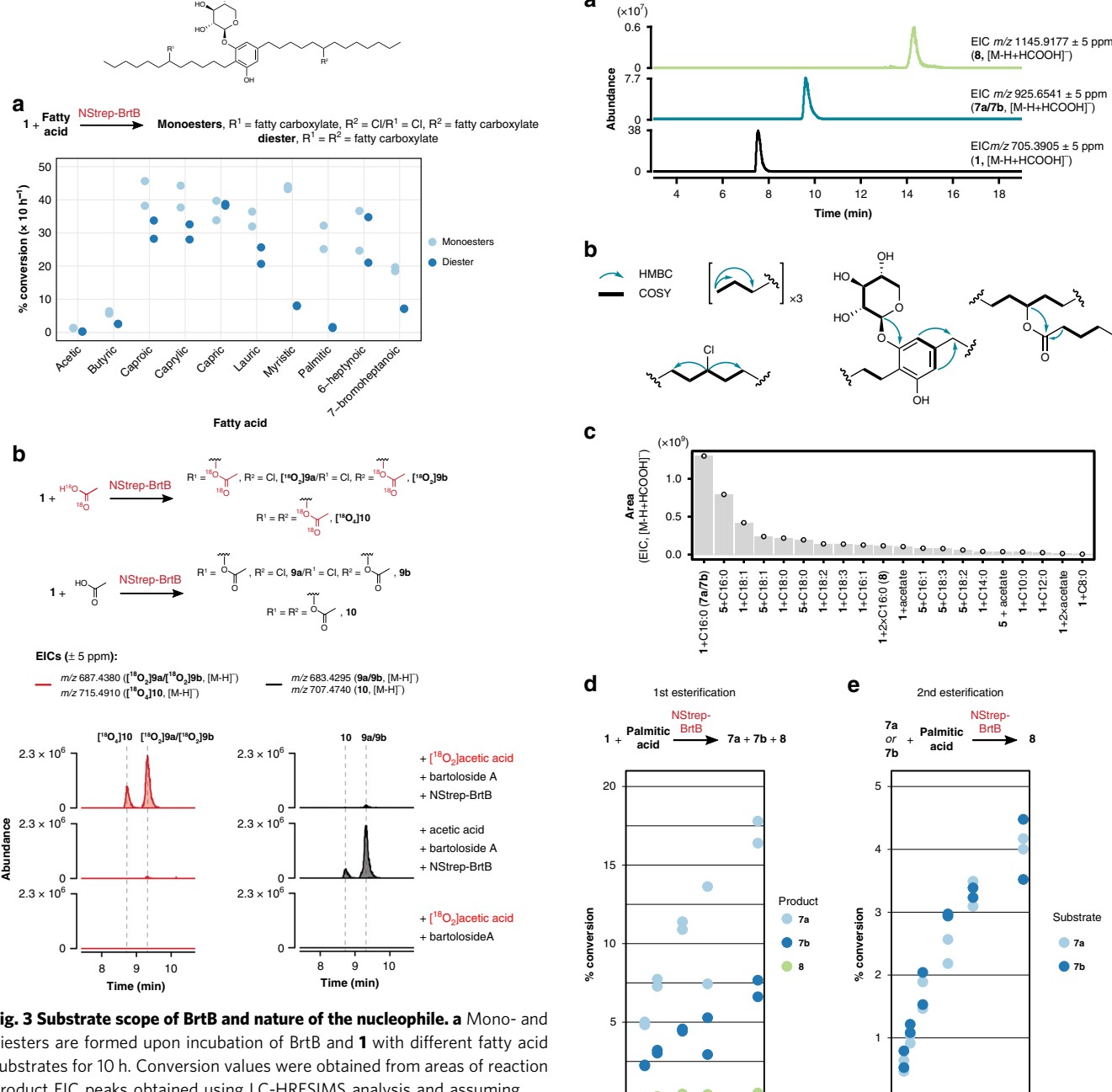

**Fig. 3 Substrate scope of BrtB and nature of the nucleophile. a** Mono- and diesters are formed upon incubation of BrtB and **1** with different fatty acid substrates for 10 h. Conversion values were obtained from areas of reaction product EIC peaks obtained using LC-HRESIMS analysis and assuming similar ionization behavior. Full reactions were carried out in duplicate and contained 0.8 μM recombinant NStrep-BrtB, 100 μM of **1** and 200 μM of fatty acid in reaction buffer. **b** LC-HRESIMS analysis of BrtB esterification reactions between **1** and [$^{18}O_2$]acetic acid or non-labeled acetic acid indicate that two $^{18}$O atoms are incorporated into the product per ester moiety.

**Fig. 4 Fatty acid-bartoloside esters are natural products. a** LC-HRESIMS analysis of an organic extract of *S. salina* LEGE 06099 suggests the presence of bartoloside A (**1**)-palmitic acid esters. **b** Key HMBC and COSY correlations obtained from 1D and 2D NMR analysis of bartoloside A palmitate **7a** purified from *S. salina* LEGE 06099 cells. **c** Bartoloside esters found in cells from a 30-day old culture of *S. salina* LEGE 06099 (as confirmed by HRESIMS/MS analysis) and their relative abundance, determined by LC-HRESIMS ($n = 1$). **d, e** Selectivity of the BrtB-catalyzed esterifications leading to bartoloside A palmitates. Shown are LC-HRESIMS analysis-derived conversions (from EIC peak areas and assuming similar ionization behavior) in assays quenched at different time points for the first esterification (**d** $n = 2$ independent assays) showing that **7a** is the preferred product and for the second esterification (**e** $n = 2$ independent assays) indicating that both **7a** and **7b** are converted into **8** at comparable rates.

Supplementary Fig. 16). To obtain further insight into the activity of BrtB, we carried out 20-min kinetic assays with **5** (with a single chlorinated position) and varying concentrations of **2**, and observed apparent $K_m$ and $V_{max}$ values of $234 \pm 41$ μM (standard error, $n = 3$) and $0.13 \pm 0.01$ μM min$^{-1}$ (standard error, $n = 3$) for the formation of **4** (Supplementary Fig. 17). Compared to haloalkane dehalogenases, BrtB has a similar apparent $K_m$ but presents a considerably higher $V_{max}$[18,19]. Regarding CylK, the single BrtB homolog characterized, there are no reported $K_m$ or $V_{max}$ values; however, it showed higher substrate conversions in 24 h assays than BrtB under similar conditions[15].

## Discussion

Our work brings to light an example, in biological chemistry, of halogenation as an intermediate step towards a non-halogenated product[20]. Cyanobacterial secondary metabolism seems to be particularly rich in such functionalization strategies (e.g., cyclophanes, curacins)[12,21]. This strategy for O–C bond formation is catalyzed by BrtB, a member of a poorly studied group of beta-propeller enzymes from which only the C–C bond forming CylK had been characterized[12,15]. These enzymes are often annotated in the GenBank as hemolysins, calcium-binding proteins, or beta-propeller proteins and contain a number of NHL repeats[12]. Several alginate C5 epimerases e.g., ref. [22], virginiamycin B lyase e.g., ref. [23] and eukaryotic CMLEs[9] have some structural homology to CylK[24]. Remarkably, both the reverse reaction of virginiamycin B lyase and the forward reaction of CMLE involve a carboxylate nucleophile[9,23]. A phylogenetic analysis of BrtB/CylK homologs (Supplementary Fig. 18) shows a large number of cyanobacterial homologs within BGCs, which most prominently encode type I polyketide synthases, fatty acid activating enzymes, non-ribosomal peptide synthetases, along with varied tailoring functionalities. Such BGCs represent clear opportunities for the discovery of natural products[12] and, in particular, the colocalization of halogenases and BrtB/CylK homologs in a variety of cyanobacterial BGCs[12] (Supplementary Fig. 18) points towards additional cryptic halogenation events. The reactivity and substrate flexibility of BrtB is also relevant for biocatalysis, owing to the ubiquity of esters in industrial and pharmaceutical chemicals, the large number of alkyl halide building blocks available and the existing precedent of using biocatalysts to generate esters at the industrial scale[25]. In particular, we envision that the ability of BrtB to esterify fatty acids of different lengths can be of use for reactions employing bio-based feedstocks such as hydrolysates from oils and fats[26]. As in the case of CylK, whether BrtB-catalyzed esterification occurs via a $S_N1$ or $S_N2$ mechanism is yet to be found[11,12]; we will focus future efforts on its structural and mechanistic characterization. Our discovery of a O–C bond forming reaction paved the way for identifying a series of natural products, the bartoloside esters, encoded by the *brt* BGC. It is unclear if these are the end-products of the clusters, since the number of glycosyltransferases in the known *brt* BGCs exceeds the observed glycosylations in the bartolosides[13,14]. The structures of the newly found bartoloside esters, with as many as 50 aliphatic methylenes, are reminiscent of certain bacterial long-chain membrane lipids (e.g., mycolic acids, heterocyst glycolipids)[27,28] and such a structural role would be in line with the high cellular abundance of bartolosides[14]. However, in light of the marked depletion of bartolosides in the presence of exogenously supplied free fatty acids, these metabolites can also represent an activation-independent fatty acid storage or scavenging mechanism[29].

## Methods

**General experimental procedures**. Fatty acids and other substrates used in feeding and in vitro assays were obtained from Fluorochem (7-bromoheptanoic, 6-heptynoic and 5-hexynoic acids), Acros Organics (palmitic and pyruvic acids, hexanamide, hexylalcohol), Alfa Aesar (caprylic and butyric acids, 1-chloro-6-phenylhexane, chlorocyclohexane, 1-chlorohexane), Thermo Fisher Scientific (acetic, caproic, capric, myristic and lauric acids), Fluka (abietic acid), Sigma-Aldrich (gallic acid, 4-chlorobutane), Santa Cruz Biotechnology ($^{18}O_2$ acetic acid) and TCI Chemicals (trans-3-hexenedioic and hexylphosphonic acids). LC-MS solvents were obtained from Thermo Fisher Scientific, Fluka and Carlo Erba. All solvents were ACS grade, except for HPLC solvents (HPLC gradient grade) and LC-MS solvents (MS-grade). Deuterated solvents for NMR were acquired from Cambridge Isotope Laboratories or Alfa-Aesar. Deuterium oxide was obtained from Sigma-Aldrich.

Luria-Bertani Lennox (LB) medium was purchased from Thermo Fisher Scientific and IPTG from NZYTech. Oligonucleotides were synthesized by StabVida. PCR reactions were carried in Veriti 96-well thermal cycler (Applied Biosystems). DNA polymerases, restriction enzymes and T4 DNA ligase were purchased from Thermo Fisher Scientific. Gel extractions of DNA fragments and restriction reactions clean ups were performed using Illustra GFX PCR DNA and Gel Band Purification kit (GE Healthcare). Plasmid purifications were performed using NZYMiniprep (NZYTech). DNA sequencing was performed by Eurofins Scientific. The Strep-Tag II kit was obtained from Novagen.

LC-HRESIMS and LC-HRESIMS/MS (HCD, Higher energy collisional dissociation) analyses were performed on an UltiMate 3000 UHPLC (Thermo Fisher Scientific) system composed of a LPG-3400SD pump, WPS-3000SL autosampler and VWD-3100 UV/VIS detector coupled to a Q Exactive Focus Hybrid Quadrupole-Orbitrap Mass Spectrometer controlled by Q Exactive Focus Tune 2.9 and Xcalibur 4.1 (Thermo Fisher Scientific). LC-HRESIMS data were obtained in Full Scan mode, with a capillary voltage of HESI set to −3.8 kV and the capillary temperature to 300 °C. Sheath gas flow rate was set to 35 units.

Optical rotations were measured in a Jasco P-2000 polarimeter controlled by SpectraManager 2.14.02 software. UV spectra were acquired on a UV-1600PC spectrometer controlled by MWAVE 1.0.20 software (VWR). Infrared spectra were acquired in a Nicolet iS5 FTIR spectrometer controlled by OMNIC 9.8.372 software (ThermoScientific). One- and two-dimensional NMR spectra were obtained in the Materials Center of the University of Porto (CEMUP) on a Bruker Avance III, 400 MHz controlled by TopSpin 3.2 or, for purified compound **7b**, on a Bruker Avance III HD, 600 MHz, equipped with a 5 mm cryoprobe and controlled by TopSpin 3.6.1. Chemical shifts are reported in parts per million (ppm) and using the residual solvent resonances as reference for $^1H$ (CDCl$_3$, 7.26 ppm) and $^{13}C$ (CDCl$_3$, 77.16 ppm). NMR data were analyzed in MNova 12.0.4.

DNA and protein concentrations were determined using a DS-11 FX Spectrophotometer/Fluorometer (DeNovix). Optical densities (600 nm) of *E. coli* cultures were determined using a Ultrospec 10 Spectrophotometer (GE Healthcare).

**Plasmids, Strains and culture conditions**. pET-51b was acquired from Novagen. *Escherichia coli* TOP10 (Life Technologies) was used for cloning and *E. coli* BL21 DE3 Rosetta (Novagen) was used for recombinant protein expression.

The cyanobacterium *Synechocystis salina* LEGE 06099 was obtained from the LEGEcc[30]. Cultures were grown in Z8 medium supplemented with 25 g L$^{-1}$ sea salt (Tropic Marin), at 25 °C, under a 14:10 h light/dark cycle and constant aeration[14]. For the feeding studies with 5-hexynoic and 6-heptynoic acids, small-scale cultures were inoculated to a final OD$_{750}$ of ~0.04–0.1 while large scale cultures (20 L) were inoculated using 1.5 L of stationary phase cultures. Feeding experiments with other fatty acids used a 3:50 inoculum of stationary phase cultures.

**Feeding experiments**. Supplementation of *S. salina* LEGE 06099 cultures (in Z8 medium, see above) with 5-hexynoic acid or 6-heptynoic acid was performed using a final concentration of 50 mg L$^{-1}$ from a 1000× concentrated solution of each acid in DMSO. The experiments were carried out in 100 mL cultures in Erlenmeyer flasks and cells were harvested after 30 days by centrifugation at $4500 \times g$, 10 min, 4 °C, rinsed with deionized H$_2$O and centrifuged again. The resulting biomass pellets were lyophilized prior to extraction with CH$_2$Cl$_2$/MeOH (2:1, v/v) at room temperature[14]. Crude extracts were analyzed by LC-HRESIMS and LC-HRESIMS/MS.

A similar procedure was used for supplementation with 7-bromoheptanoic, caprylic, butyric, lauric and palmitic acids, but a final concentration of each substrate of 0.4 mM was used. Cells were harvested after three days of exposure, and were extracted with CH$_2$Cl$_2$/MeOH (2:1, v/v) at room temperature, without prior lyophilization. The resulting crude extracts were analyzed by LC-HRESIMS.

In order to obtain sufficient biomass to isolate bartoloside esters formed upon supplementation, a 20 L culture of *S. salina* LEGE 06099 was prepared in Z8 medium supplemented with 6-heptynoic acid (50 mg L$^{-1}$), as described above. After a 45 days growth period, the biomass was harvested by centrifugation ($4500 \times g$, 10 min, 4 °C), washed with deionized water and centrifuged again. The corresponding pellet was freeze-dried and stored at −20 °C until further use.

**LC-HRESIMS analysis**. For LC-HRESIMS analyses, separation was performed in a Luna C18 column (100 × 3 mm, 3 μm, 100 Å, Phenomenex). Mixtures of MeOH/H$_2$O 1:1 (v/v) with 0.1% formic acid (eluent A) and IPA with 0.1% formic acid (eluent B) were used as mobile phase, with a flow rate of 0.4 mL min$^{-1}$. Depending on the sample to be analyzed, different elution programs and injection volumes/concentrations were used, as detailed below.

Crude extracts and flash chromatography fractions obtained from the initial feeding experiments were separated (10 μL of a 1 mg mL$^{-1}$ solution were injected) using a gradient from 9:1 to 3:7 eluent A/eluent B in 10 min and held for 7 min before returning to the initial conditions.

Crude extracts obtained from feeding with 7-bromoheptanoic, caprylic, butyric, lauric and palmitic acids were separated using a gradient from 9:1 to 3:7 eluent A/eluent B in 10 min and held for 12 min before returning to the initial conditions.

Crude extracts of biomass from cultures that were not supplemented with fatty acids were analyzed by LC-HRESIMS using a gradient from at 9:1 to 1:4 eluent A/eluent B in 3 min and held for 28 min before returning to initial conditions. Fractions obtained from semipreparative HPLC during purification of natural bartoloside esters were analyzed (10 μL at 1 mg mL$^{-1}$ per injection) using a similar program, but with an isocratic step of only 24 min. Semipreparative HPLC

subfractions that were analyzed by LC-HRESIMS/MS were separated (10 μL at 0.5 mg mL$^{-1}$ injected) using a gradient from 9:1 to 7:13 eluent A/eluent B in 5 min, increasing to 3:7 eluent A/eluent B over 15 min and held for 5 min before returning to the initial conditions.

Analysis of enzymatic reaction samples was carried out by injecting 10 μL of supernatant from the methanol/acetonitrile-quenched reaction mixture (see Enzymatic assays). Separation involved an isocratic step of 9:1 eluent A/eluent B over 2 min, followed by a linear gradient to 7:13 eluent A/eluent B over 3 min and held for 10 min, followed by a linear gradient to 3:17 eluent A/eluent B over 3 min and held for 8 min before returning to the initial conditions. For the activity dependence on temperature, esterification in D$_2$O assay and assays for the determination of kinetic parameters, separation employed an isocratic step of 9:1 eluent A/eluent B over 2 min, followed by a linear gradient to 7:13 eluent A/eluent and held for 7 min before returning to the initial conditions. A longer program that allowed for the separation of monoesters **7a/7b** was used to analyze the assays designed to study the selectivity of the esterification of **1** and palmitic acid. It consisted of a flow rate of 0.6 mL min$^{-1}$ and an isocratic step of 9:1 eluent A/eluent B over 2 min, followed by a linear gradient to 9:11 eluent A/eluent B over 2 min, held for 38 min, then by a linear gradient to 3:17 eluent A/eluent B over 5 min and was held for 8 min before returning to the initial conditions.

**MS/MS analysis**. MS/MS parameters for the LC-HRESIMS/MS analysis of crude extracts, HPLC fractions and enzymatic assays were: resolution of 35000, with a 1 *m/z* isolation window, a loop count of 3, AGC target of 5 × 10$^4$ and collision energy of 35 eV.

HRESIMS/MS analysis of purified compounds **3** and **4** was performed by direct injection (0.1 mg mL$^{-1}$ solutions) into the spectrometer, using a resolution of 35000, a 1 *m/z* isolation window, a loop count of 3 and an AGC target of 5 × 10$^4$. Stepped collision energies of 35, 40, and 45 eV were applied. To obtain structural information regarding the dialkylresorcinol moiety, in-source-formed species were selected for ddMS$^2$ events[14]. For purified compounds **3** and **4**, the in-source collision induced dissociation (CID) energy was set to 90 and 65 eV, respectively (to isolate species corresponding to loss of xylosyl, C$_7$H$_9$O and C$_7$H$_9$O$_2$ for **3** as well as loss of xylosyl and C$_7$H$_9$O for **4**), resolution of 35,000, with 1 *m/z* isolation window, loop count of 1 and AGC target of 5 × 10$^4$. CID energy was set to 55 eV. For purified compounds **7a** and **7b** the in-source collision energy was set to 90 eV (to isolate a species corresponding to the loss of xylosyl and HCl).

**Isolation of bartoloside esters 3 and 4**. The freeze-dried biomass (5.6 g, d.w.) from a 20 L culture of *S. salina* LEGE 06099 in Z8 medium supplemented with 6-heptynoic acid (details in Feeding Experiments section) was extracted by repeated percolation using a mixture of CH$_2$Cl$_2$/MeOH (2:1, v/v) at room temperature. The resulting crude extract (754.6 mg) was fractionated by normal phase flash chromatography (Si gel 60, 0.040-0.063 mm, Macherey-Nagel) using a gradient of increasing polarity from hexane to EtOAc to MeOH, yielding eight fractions (1-8). Fraction 5 (102.2 mg), eluting with a 2:3 mixture of EtOAc/hexane was further separated by RP-semipreparative HPLC with a ACE C18 column (100 Å pore size, 250 × 10 mm, 5 μm) (ACE). Mobile phase A was ddH$_2$O and mobile phase B was MeCN (aq). The LC method used a constant flow rate of 3 mL min$^{-1}$ with an isocratic step 8% A, 92% B over 25 min followed by a gradient to 100% B over 2 min and held for 15 min before returning to the initial conditions. This procedure afforded 10 subfractions. Fractions 5.6 (compound **3**, 48.0 mg, t$_R$ = 13.5–15.0 min) and 5.10 (compound **4**, 4.3 mg, t$_R$ = 24.0–25.0 min) were spectroscopically pure (~95%, $^1$H NMR), showing signals characteristic of bartolosides[14].

*bartoloside A-17,29-diyl bis(hept-6-ynoate) (3) (orange oil):* [α]$_D^{25}$ −5.0 (c 0.60, MeOH); IR (thin film) ν$_{max}$ 3398, 3308, 2923, 2857, 1726, 1703, 1590, 1428, 1052, 1033, 1017, 627 cm$^{-1}$; UV (MeOH) λ$_{max}$ (log ε) 217 (3.2), 221 (3.2), 273 (3.0); $^1$H and $^{13}$C NMR data, Supplementary Table 1; HRESIMS *m/z* 839.5676 [M-H]$^-$ (calcd for C$_{50}$H$_{79}$O$_{10}$, 839.5679).

*bartoloside G-17-yl hept-6-ynoate (4) (light orange oil):* [α]$_D^{25}$ −4.6 (c 0.65, MeOH); IR (thin film) ν$_{max}$ 3410, 3313, 2923, 2855, 1731, 1704, 1590, 1428, 1054, 1033, 1014, 627 cm$^{-1}$; UV (MeOH) λ$_{max}$ (log ε) 216 (3.1), 221 (3.1), 273 (2.8); $^1$H and $^{13}$C NMR data, Supplementary Table 2; HRESIMS *m/z* 715.5153 [M-H]$^-$ (calcd for C$_{43}$H$_{71}$O$_8$, 715.5154).

**Isolation of bartoloside esters 7a and 7b**. Freeze-dried biomass (5.3 g, d.w.) from a 30-day old, 20 L culture of *S. salina* LEGE 06099 in Z8 medium that was not supplemented with any fatty acid (instead, 0.5% DMSO was added) was extracted as detailed above for the isolation of compounds **3** and **4**. The resulting organic extract (574.3 mg) was fractionated by normal phase flash chromatography (Si gel 60, 0.040–0.063 mm, Macherey-Nagel). An increasing polarity gradient from hexane to EtOAc to MeOH was used for elution. The collected fractions were pooled according to their TLC profiles, affording 23 sub-fractions (1-23). Sub-fraction 15, eluting with 2:3 and 1:1 EtOAc/hexane, contained *m/z* 879.648 (**7a/7b**, [M-H]$^-$), as verified by LC-HRESIMS analysis, and was further fractionated by RP-semipreparative HPLC (ACE C18, 250 × 10 mm, 5 μm, 100 Å, ACE), using 98% MeOH (aq) isocratically over 65 min and with a flow rate of 3 mL min$^{-1}$. $^1$H NMR analysis showed that a minor bartoloside co-eluted (likely compound **7b**), and thus further separation was necessary. Fraction 15.7 (t$_R$ = 32.1–34.5 min) was reinjected

into the RP-semipreparative system using the above-mentioned column and further separated with 98% MeCN (aq) to yield metabolite **7a** (3.1 mg, t$_R$ ~122 min). $^1$H NMR analysis indicated that the compound was pure (>95%) and showed characteristic resonances for bartolosides[14].

*bartoloside A-17-yl palmitate (7a) (white glassy solid):* [α]$_D^{25}$ − 9.2 (c 0.42, MeOH); IR (thin film) ν$_{max}$ 3417, 2923, 2854, 2360, 2340, 1730, 1704, 1592, 1463, 1455, 1430, 1159, 1048, 1033 cm$^{-1}$; UV (MeOH) λ$_{max}$ (log ε) 217 (3.3), 222 (3.3), 273 (2.9); $^1$H and $^{13}$C NMR data, Supplementary Table 3; HRESIMS *m/z* 879.6499 [M-H]$^-$ (calcd for C$_{52}$H$_{92}$O$_8$Cl, 879.6486).

This procedure also yielded partially purified metabolite **7b** (1.2 mg, t$_R$ ~126 min). $^1$H NMR analysis indicated that the compound was ~80% pure with the impurity being **7a**. The partially purified **7b** showed $^1$H, $^{13}$C and HRESIMS data ($^1$H and $^{13}$C NMR data, Supplementary Table 4; HRESIMS *m/z* 879.6497 [M-H]$^-$ (calcd for C$_{52}$H$_{92}$O$_8$Cl, 879.6486)) were very similar to those of **7a**. Confirmation of the identity of **7b** in this sample came from comparison of $^1$H NMR and HRESIMS/MS data with **1** and **7a** (Supplementary Note 1).

**Cloning, expression, and purification of NStrep-BrtB**. Genomic DNA from *S. salina* LEGE 06099 was extracted using a CTAB-chloroform/isoamyl alcohol-based protocol[31] and was used for the PCR amplification of *brtB* using primers BrtB-NStrep-SalI-F (5′ AATGTCGACATGGCCAATCCCTTCG 3′, forward) and BrtB-NStrep-SacI-R (5′ ATATGAGCTCCTAGTAGCCGTACCCG 3′, reverse), which contained restriction sites for SalI and SacI, respectively (underlined in the sequences). The reaction was prepared in a volume of 20 μL, containing 1 × Phusion HF Buffer, 400 μM dNTPs, 0.5 μM of each primer, 0.4U Phusion polymerase (Thermo Scientific) and 2 μL of template gDNA. The PCR cycling program consisted in a two-step protocol with extension at 72 °C for 1.5 min. PCR products were gel purified, restriction digested using SalI and SacI (Thermo Scientific) for 1.5 h at 37 °C and gel purified again. Digests were ligated into linearized StrepTag-containing pET-51b (+), using T4 DNA Ligase at 16 °C overnight. This affinity tag was chosen because functional, StrepTag-fused CylK had been successfully purified by Balskus and co-workers[12,15]. Competent *E. coli* TOP10 cells were transformed with a 5 μL ligation sample by incubating the mixture on ice for 30 min, heat shocking at 42 °C for 45 s and recovering on ice for 2 min. Then, 250 μL of warm S.O.C. medium was added, the mixture was incubated at 37 °C for 1 h and 100 μL of cells was plated on LB agar supplemented with 100 μg mL$^{-1}$ ampicillin. The plate was incubated overnight at 37 °C. Individual colonies were inoculated into 5 mL of LB supplemented with 100 μg mL$^{-1}$ ampicillin and grown overnight at 37 °C with 175 rpm shaking. The purified plasmids were sequenced to verify construct identity and amplification fidelity. pET-51b-NStrep-*brtB* was transformed into chemically competent *E. coli* BL21 Rosetta (DE3) by incubating the plasmid and cells on ice for 5 min, applying a heat-shock at 42 °C for 45 s and recovering for 2 min on ice. Then, 80 μL of warm S.O.C. medium was added, the mixture was incubated at 37 °C for 1 h and 70 μL of cells was plated on LB agar supplemented with 100 μg mL$^{-1}$ ampicillin and 34 μg mL$^{-1}$ chloramphenicol. Individual colonies were inoculated into 5 mL of LB supplemented with 100 μg mL$^{-1}$ ampicillin and 34 μg mL$^{-1}$ chloramphenicol and grown overnight at 37 °C with 175 rpm shaking. The resulting transformants were frozen in LB/glycerol 1:1 (v/v) stocks and stored at −80 °C.

An initial protein preparation was made starting from the glycerol stock of *E. coli* BL21 Rosetta (DE3) harboring the pET-51b-NStrep-*brtB* plasmid was used to prepare a 50 mL starter culture in LB medium supplemented with 100 μg mL$^{-1}$ ampicillin and 34 μg mL$^{-1}$ chloramphenicol. The starter culture was incubated overnight at 37 °C with 175 rpm shaking. The overnight culture was used to inoculate (1:100 dilution) 2 L of LB medium supplemented with 100 μg mL$^{-1}$ ampicillin and 34 μg mL$^{-1}$ chloramphenicol. The culture was incubated at 37 °C with 190 rpm shaking. At OD$_{600}$ = 0.4-0.5 the culture was transferred to 15 °C and orbital shaking decreased to 175 rpm. Once the culture reached OD$_{600}$ = 0.8-0.9, protein expression was induced with 500 μM IPTG and the culture was supplemented with 5 mM CaCl$_2$[15]. After four additional hours of incubation at 15 °C with 175 rpm shaking, the cells were harvested by centrifugation (3000 × *g*, 10 min, 4 °C), washed with Phosphate-buffered saline (PBS) and re-centrifuged. The pellet was flash frozen in liquid nitrogen and stored at −80 °C until it was resuspended in lysis buffer (50 mM Tris-HCl, 500 mM NaCl, 20 mM CaCl$_2$, 10 mM MgCl$_2$, 1 mM EDTA, pH 8.0) to which Pierce Protease Inhibitor Mini Tablets EDTA-free were added according to the manufacturer's instructions (Thermo Scientific). The cells were lysed in a cell disruptor (CF Range, Constant Sytems Ltd) by a single cycle at 20000 psi and the lysate was clarified by centrifugation (19500 × *g*, 30 min, 4 °C). A pre-packed 1 mL StrepTactin Superflow column (Novagen) was equilibrated with 2 mL of lysis buffer at 4 °C for 45 min and the clarified lysate was transferred to the column, discarding the flow-through. The column was washed with 5 mL lysis buffer and the protein was eluted in multiple 500 μL fractions using elution buffer (100 mM Tris-HCl, 500 mM NaCl, 20 mM CaCl$_2$, 10 mM MgCl$_2$, 1 mM EDTA, 2.5 mM desthiobiotin, pH 8.0). The collected fractions were analyzed by SDS-PAGE (4-20% Mini-PROTEAN TGX precast gel, BIO-RAD). Fractions containing the desired protein (~91.4 kDa) were pooled and desalted using PD-10 desalting columns (GE Healthcare). The columns were equilibrated with storage buffer (50 mM Tris-HCl, 50 mM NaCl, 10 mM MgCl$_2$, 10 mM CaCl$_2$ and 10% glycerol, pH 8.0) and the protein sample was loaded into the columns (up to 2.5 mL per column) and eluted with 3.5 mL storage buffer. The sample was concentrated to

34.6 μM using Pierce™ Protein Concentrator tubes, 10 K MWCO (Thermo Scientific) through two consecutive 15 min centrifugations ($8000 \times g$, 4 °C). The concentrated protein was frozen in liquid nitrogen as pellets and stored at –80 °C. This procedure yielded 0.3 mg of NStrep-BrtB per L of E. coli culture (Supplementary Fig. 19).

A second protein preparation was performed after inoculating a 100 mL starter culture of the above-mentioned strain which was then used to inoculate (1:100 dilution) 4 L of LB medium supplemented with 100 μg mL$^{-1}$ ampicillin and 34 μg mL$^{-1}$ chloramphenicol. The conditions and procedures applied were repeated with exception of the following steps. Pellets were resuspended in lysis buffer (20 mM HEPES, 500 mM NaCl, 20 mM CaCl$_2$, 10 mM MgCl$_2$, pH 8.0) to which Pierce Protease Inhibitor Tablets EDTA-free were added. The new affinity chromatography used 6 mL of a 50% Strep-Tactin Superflow resin and the column equilibrated with 20 mL lysis buffer. The clarified lysate was transferred to the column, discarding the flow-through. The column was washed with 20 mL lysis buffer and protein was eluted in multiple 1.5 mL fractions using elution buffer (lysis buffer with 2.5 mM desthiobiotin, pH 8.0). The desired fractions were pooled, according to the SDS-PAGE results, and dialyzed overnight against 4 L of dialysis buffer (20 mM HEPES, 50 mM NaCl, 10 mM MgCl$_2$, 10 mM CaCl$_2$, 10% glycerol, pH 8.0). The sample was concentrated to 28.6 μM using Pierce™ Protein Concentrator tubes, 10 K MWCO (Thermo Scientific) through two consecutive 15 min centrifugations ($8000 \times g$, 4 °C). The concentrated protein was frozen in liquid nitrogen as pellets and stored at −80 °C. This procedure yielded 0.5 mg of NStrep-BrtB per L of E. coli culture (Supplementary Fig. 19).

**Enzymatic assays**. In vitro enzymatic assays to validate BrtB function were carried out in Eppendorf tubes to a final volume of 100 μL in reaction buffer (50 mM Tris-HCl, 50 mM NaCl, 10 mM MgCl$_2$, 5 mM CaCl$_2$, pH 8.0). Final concentrations of each substrate were 1 μM for NStrep-BrtB, 100 μM for bartoloside A (**1**) and 200 μM for both 6-heptynoic (**2**) and palmitic acids and the reactions were incubated at 37 °C with 180 rpm shaking. Aliquots (30 μL) were collected at 0, 6 and 24 h and quenched with 60 μL of a cold mixture of MeOH/MeCN (1:1, v/v), vortexed immediately and incubated in ice for 10 min. Quenched reactions were then centrifuged at 17 000 × g, 4 °C, 10 min) and analyzed by LC-HRESIMS/MS.

In vitro enzymatic assays for testing the influence of temperature on activity and for substrate scope studies were carried out in Eppendorf tubes to a final volume of 100 μL in reaction buffer (20 mM HEPES, 50 mM NaCl, 10 mM MgCl$_2$, 5 mM CaCl$_2$, pH 8.0), over 24 h (temperature) and 10 h (substrate scope). Experiments using $^{18}O_2$-labeled and unlabeled acetic acid and deuterium oxide (D$_2$O) were carried in similar conditions but with a final volume of 50 μL and over 24 h and 16 h, respectively. For the D$_2$O experiments, the reaction buffer (20 mM HEPES, 50 mM NaCl, 10 mM MgCl$_2$, 5 mM CaCl$_2$, pH 8.0) was fully evaporated in a centrifugal evaporator at 37 °C and dissolved in D$_2$O. Final concentrations of each substrate were 0.8 μM (1 μM for D$_2$O and $^{18}O_2$ experiments) for NStrep-BrtB, 100 μM for bartoloside A (**1**) and 200 μM for the substrates (100 μM for substrate scope reactions in which monochlorinated alkyl halides were used as electrophilic substrates) and the reactions were incubated at 37 °C (and additionally at 15 °C, room temperature—~25 °C—and 55 °C for the temperature experiment) with 180 rpm shaking. Aliquots (30 μL) were collected and quenched with 150 μL (60 μL for temperature dependence and $^{18}O_2$ experiments) of a cold mixture of MeOH/MeCN (1:1, v/v), vortexed immediately and incubated in ice for 10 min. We found that quenching with five volumes of the organic mixture led to improved signal in LC-HRESIMS analysis for the bartoloside esters with the longer fatty acids, when compared to the two volumes used in the initial assays with BrtB, likely due to solubility issues. Quenched reactions were then centrifuged at 17,000 × g, 4 °C, 10 min) and analyzed by LC-HRESIMS/MS. Percent (%) conversion was determined by integrating EIC peaks for each species and assumed similar ionization behavior.

Assays for the selectivity of the first and second esterifications of **1** and palmitic acid were carried in 250 μL reaction buffer (20 mM HEPES, 50 mM NaCl, 10 mM MgCl$_2$, 5 mM CaCl$_2$, pH 8.0). For the first esterification, final concentrations of 1 μM for NStrep-BrtB, 100 μM for bartoloside A (**1**) and 200 μM for palmitic acid were used and for the second 100 μM for the substrates (palmitic acid, **7a**, **7b**, equimolar mixture of **7a/7b**) were used instead, so as to maintain the stoichiometric ratio. Reactions were incubated at 37 °C with 180 rpm shaking. The equimolar mixture of **7a/7b** was prepared considering $^1$H NMR-derived purity estimations. Aliquots (30 μL) were collected and quenched with 150 μL of a cold mixture of MeOH/MeCN (1:1, v/v), vortexed immediately and incubated in ice for 10 min. Quenched reactions were then centrifuged at 17,000 × g, 4 °C, 10 min) and analyzed by LC-HRESIMS/MS. Percent (%) conversion was determined by integrating EIC peaks for each species and assumed similar ionization behavior.

**Determination of kinetic parameters for NStrep-BrtB**. For the determination of apparent $K_m$ and $V_{max}$ parameters for NStrep-BrtB, the two protein preparations that had been obtained were pooled in a 1:1 ratio (Supplementary Fig. 19). Assays were carried in 120 μL reaction buffer (20 mM HEPES, 50 mM NaCl, 10 mM MgCl$_2$, 5 mM CaCl$_2$, pH 8.0). Bartoloside G (**5**), obtained previously[14] at a final concentration of 100 μM was used in the assays because a single site is available for esterification. A range of 6-heptynoic acid (**2**) concentrations (corresponding

to 5, 25, 50, 100, 175, 250, 500, 750, and 1000 μM and prepared from a 100 mM stock solution) were used to determine the initial reaction rates. Reactions were incubated at 37 °C and aliquots (20 μL) were collected at different time points (5, 10, 20, 30, and 60 min) and quenched with 60 μL of a cold mixture of MeOH/MeCN (1:1, v/v), vortexed immediately and incubated in ice for 10 min. Quenched reactions were then centrifuged at 17 000 × g, 4 °C, 10 min) and analyzed by LC-HRESIMS/MS.

A calibration curve of purified compound **4** was generated (EIC peak area = $1.57 \times 10^{-8} \times$ [**4**, μM] $-3.63 \times 10^{-2}$) using dilutions from a 10 mM standard of compound **4** (0.005, 0.01, 0.1, 0.5, 1, 5, and 10 μM) and considering the integration of the EIC peak for m/z 761.522 ([M-H + HCOOH]$^-$, **4**). The initial reaction rates ($V_0$) were calculated from each linear regression slope for the formation of **4** over time (considering only time points 5, 10, and 20 min) and were then used to calculate apparent $K_m$ and $V_{max}$ values using R[32].

**Bioactivity assays**. Cytotoxicity assays against human immortalized cells: Compounds **3** and **4** were tested for cytotoxicity against the HCT116 colon colorectal carcinoma, HT-29 colon colorectal adenocarcinoma and hCMEC/D3 blood-brain barrier cell lines and compound **7a** was tested in the HT-29 cell line. All cell lines were sub-cultivated and grown in supplemented medium as recommended by the providers. All HCT116 cells were maintained in McCoy's 5 A modified medium while HT-29 and hCMEC/D3 cell lines were maintained in Dulbecco's Modified Eagle Medium (DMEM). All media were also supplemented with 10% fetal bovine serum (Biochrom), 1% of penicillin/streptomycin (Biochrom), and 0.1% of Amphotericin B (GE Healthcare), and all cell lines incubated at 37 °C in 5% CO$_2$. For the cytotoxicity assays, cells were seeded at $3.3 \times 10^4$ cells mL$^{-1}$ in 96-well plates, exposed to compounds **3**, **4**, or **7a** after 24 h and kept in the incubator for 24 and 48 h. All compounds were prepared in 100× concentrated stock solutions dissolved in DMSO. A volume of 1 μL of each stock solution was added to each well to a final concentration of 0.1% DMSO; negative control wells received solvent only; positive controls used a solution of staurosporine dissolved in DMSO, added to a final concentration of 1 μM. After the exposure, 3-(4,5-dimethylthiazol-2-yl)-2,5-diphenyltetrazolium bromide (MTT) was added at a final concentration of 0.05 mg mL$^{-1}$ per well for 3 h. The formation of formazan crystals was visually evaluated by microscopy, before being dissolved in DMSO and the absorbance in each well measured at 550 nm on a Synergy HT Microplate Reader (Biotek).

Antimicrobial agar disk diffusion assays: Solutions of 0.5 mg mL$^{-1}$ of compounds **3**, **4**, and **7a**, prepared in DMSO were tested against a Gram positive bacterial strain (Staphylococcus aureus ATCC 29213), Gram-negative bacterial strains (Escherichia coli ATCC 25922 and Salmonella typhimurium ATCC 25241) and a yeast strain (Candida albicans ATCC 10231). The bacteria were grown in Mueller-Hinton agar (MH—BioKar diagnostics) from stock cultures and incubated at 37 °C. The yeast was grown in Sabouraud Dextrose Agar (BioKar diagnostics). For the antimicrobial screening, a disk diffusion test was carried out. Bacterial colonies were picked from overnight cultures in MH and suspended in LB liquid medium, the turbidity adjusted to OD$_{600}$ = 0.090–0.110 (0.5 McFarland standard) and the MH plates seeded with the resulting inoculum. Blank disks (6 mm in diameter, Oxoid) were placed in the inoculated plates and impregnated with 15 μL of a 0.5 mg mL$^{-1}$ solution of each compound. The plates were left for 30 min at room temperature and then incubated overnight at 37 °C. After 24 h, the plates were checked for inhibition halos, indicative of antimicrobial activity. A disk with 15 μL DMSO was used as negative control.

**Phylogenetic analysis**. Amino acid sequences of the BrtB homologs with lowest e-value (BlastP) were retrieved from the NCBI database (232 homologs retrieved). Their amino acidic sequences were aligned, together with those of a set of the distantly related virginiamycin lyases and alginate C5 epimerases and with the sequence of BrtB, using MUSCLE, from within Geneious R11 (Biomatters). The resulting alignment (with a total of 239 sequences) was trimmed to its core region and contained 1558 positions. FastTree 2.1.5[33] (from within Geneious R11) was used to compute an approximately-maximum-likelihood phylogenetic tree, using pseudocounts and 1000 rate categories of sites.

**Availability of materials**. Plasmids and strains are available from the corresponding author upon reasonable request.

**Reporting summary**. Further information on research design is available in the Nature Research Reporting Summary linked to this article.

## Data availability
The data that support the findings of this study are available from the corresponding author upon reasonable request. The source data underlying Supplementary Figs. 17 and 19 are provided as a Source Data file.

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

## Acknowledgements

We acknowledge funding by the European Research Council, through a Starting Grant (Grant Agreement 759840) to P.N.L., and by Fundação para a Ciência e Tecnologia (FCT) through project PTDC/BIA-BQM/29710/2017 and grant IF/01358/2014 to P.N.L. The work was also partially supported by Strategic Funding UIDB/04423/2020 and UIDP/04423/2020 by FCT and the European Regional Development Fund, as part of the program PT2020. We thank Emily Balskus (Department of Chemistry and Chemical Biology, Harvard University, Cambridge, MA, USA) for helpful discussions and Ralph Urbatzka (CIIMAR, University of Porto, Porto, Portugal) for help with cytotoxicity assays.

## Author contributions

J.P.A.R. and P.N.L conceived the project, J.P.A.R., S.A.C.F., and M.L.S. performed experimental work, J.P.A.R. and P.N.L. wrote the manuscript with input from S.A.C.F. and M.L.S.

## Competing interests

The authors declare no competing interests.
