## [Peer Review File · Nature Communications]

Reviewers' comments:

Reviewer #1 (Remarks to the Author):

In this manuscript, Reis et al report characterization of BrtB as an O-alkylating enzyme, which allowed the production of a series of esters from chlorinated bartoloside and different fatty acids. In general, the paper is well written and much of the experiment data are very solid. I would suggest that the authors address the following major issues before its acceptance for publication. Currently, I am not very convinced about the significance of this study.

1) Ester formation involving a carboxylate nucleophile is NOT as unusual as the authors claimed. There are several other examples besides NosN, and the most prominent one is the methyl ester formation via methylation of a carboxylate (I feel surprised that the authors appeared to not even think about this).

2) Although enzyme activity was shown in vivo and in vitro, there are no further analysis of the enzyme selectivity and specificity. Since the reaction occurs on both C17 and C29 of 1, and two monoesters were observed in the reaction, which site is more preferential in the reaction? What is the ratio of 7a and 7b? Why only 7a was purified and structurally characterized? More importantly, I believe kinetic studies should be performed with both monoesters (ideally also with 1) to provide more insights into the reaction path.

3) I am not very comfortable with the proposal that fatty acid-bartoloside esters are the end-products of bartoloside. Are all the enzymes encoded in the gene cluster fully characterized? Could the author also discuss the functions and/or biological activities of these fatty acid esters?

4) What is the point of their phylogenetic analysis? Does it just show the widespread occurrence of BrtB homologues? The authors may, ideally, show the function of one or two of these homologues. I would also suggest that the author move this section to the end of the manuscript.

Reviewer #2 (Remarks to the Author):

The manuscript "BrtB is a novel O-alkylating enzyme that generates fatty acid-bartoloside esters" describes a very interesting new enzyme activity. An enzyme that catalyses the Fischer (not Fisher) Speier esterification. The analytics of the enzyme catalysed reactions are thorough and convincing but the data on the enzyme are missing. Essentially one only learns that it has been cloned and the gene does express. The authors need to provide the biochemical characterisation of the enzyme too. Without this the study is incomplete.

Reviewer #3 (Remarks to the Author):

The paper describes the discovery of an unusual O-alkylating enzyme BrtB, suggested to function through nucleophilic attack of a fatty acid carboxylate on halogenated bartolosides to produce a wide range of new natural product esters. Overall this is a nice story/discovery. BrtB is a novel enzyme and there is good characterisation/evidence of the products formed in vivo and from in vitro reactions. They also show the the enzyme is fairly promiscuous with respect to the fatty acid substrate. In light of this I believe the paper would be of interest to those in the biosynthesis and enzymology field. However, there are a number of issues that would need to be addressed before publication:

The introduction suggests that there is only one other example in nature where a carboxylate acts as a nucleophile for ester formation (Fig 1a). Actually there are a number O-methyltransferases where a carboxylate substrate acts as a nucleophile with SAM in a SN2 reaction to form methyl esters which is directly analogues to the proposed reactivity of BrtB.

<https://www.ncbi.nlm.nih.gov/pmc/articles/PMC167163/>

<http://www.jbc.org/content/287/44/37010.full.pdf>

<https://aac.asm.org/content/58/2/950>

The section in the middle of the paper discussing bioinformatics analysis and potential BGCs containing BrtB like enzymes is too long and does not come up with any useful hypothesis regarding what other similar esters might be out there and what their role might be in nature.

They assume the reaction proceeds by an SN2 mechanism similar to haloalkane dehalogenases but provide no evidence to support this. They should determine the absolute configuration of the substrates and products, so that they can establish if the reaction proceeds with overall retention or inversion of stereochemistry. Inversion would support the idea of direct attack of the fatty acid carboxylate onto the alkylhalide and retention could indicate possible covalent-catalysis proceeding via initial attack of active site nucleophile (e.g. Asp or Glu as in haloalkane dehalogenases).

They use a lot of unsure language particularly in the conclusion – serendipitous, seems, suggests, likely, speculate, might, tempting. Whilst, I appreciate it is important to not be too speculative or overstate the case, the language does give the impression that the authors are not convinced by their conclusions.

The paper suggests BrtB might be useful as an industrial biocatalyst, but does not provide any suggestions or examples of how such applications might be realised.

The Figures are cluttered. For example, Fig 2 bottom right hand corner looks like it is linked to part d, but it is linked to c. Fig 2d is far too small (illegible) and provides little information. It should be removed with an expanded and more detailed version added to SI.

BrTB is a novel O-alkylating enzyme that generates fatty acid-bartoloside esters

Response to Reviewers' comments (responses in blue color):

Reviewer #1 (Remarks to the Author):

In this manuscript, Reis et al report characterization of BrTB as an O-alkylating enzyme, which allowed the production of a series of esters from chlorinated bartoloside and different fatty acids. In general, the paper is well written and much of the experiment data are very solid. I would suggest that the authors address the following major issues before its acceptance for publication. Currently, I am not very convinced about the significance of this study.

1) Ester formation involving a carboxylate nucleophile is NOT as unusual as the authors claimed. There are several other examples besides NosN, and the most prominent one is the methyl ester formation via methylation of a carboxylate (I feel surprised that the authors appeared to not even think about this).

Unfortunately and, as the referee suggests, we did not even consider the methylation reaction. We apologize for the oversight and thank the referee for alerting us to this issue and preventing us from making an unsubstantiated claim.

We now include this and other examples and tone down on some of our statements accordingly, while emphasizing the actual novelty of our findings, i.e. that in the newly described reactivity, the nucleophile is a non-activated fatty acid and the electrophile is an alkyl halide.

2) Although enzyme activity was shown in vivo and in vitro, there are no further analysis of the enzyme selectivity and specificity. Since the reaction occurs on both C17 and C29 of 1, and two monoesters were observed in the reaction, which site is more preferential in the reaction? What is the ratio of 7a and 7b? Why only 7a was purified and structurally characterized? More importantly, I believe kinetic studies should be performed with both monoesters (ideally also with 1) to provide more insights into the reaction path.

We now provide insight into enzyme selectivity for the first and also for the second esterification steps. This involved purifying the minor monoester 7b, which was demanding due to the lack of sufficient chromatographic separation. We ended up being able to partially purify 7b (~80%, ¹H NMR, the impurity being 7a) and characterize it by NMR. In fact, our previous purification of the 7a metabolite had also some 7b as impurity (~5-10%). These two purified monoesters served as standards in the LC-MS assays to test for selectivity in the first esterification step. Using the ¹H NMR-derived purity estimations, we were able to prepare equimolar mixtures of 7a and 7b in order to test the selectivity of the enzyme in the second esterification step, in competition assays, but also using 7a and 7b individually, in kinetic studies as suggested by the referee. The new data is discussed in the manuscript and provided in Fig. 3a and in the SI.

3) I am not very comfortable with the proposal that fatty acid-bartoloside esters are the end-products of bartoloside. Are all the enzymes encoded in the gene cluster fully characterized? Could the author also discuss the functions and/or biological activities of these fatty acid esters?

With the characterization of BrTB, we now only eventually miss the role of one of the glycosyltransferases; the reviewer is right in mentioning that we are not entirely sure that the esters we just found are the end-products of the brt clusters. We now mention this in the text and the possibility that these esters could be further glycosylated, although we never found any m/z values that could indicate such a transformation. Another possibility is that two glycosyltransferases are involved in the installation of the O-glycoside, since the brt cluster encoding the diglycosylated bartolosides has three glycosyltransferases, while the one encoding the monoglycosylated bartolosides has two glycosyltransferases. Although we find it likely that bartoloside esters are, in fact, the end-products of the brt cluster, we do not have proof or indication for the role of the additional glycosyltransferase. As such, we have now removed any statements indicating that bartoloside esters are the end-products of the brt clusters.

Regarding biological activities, we have tested the compounds in anticancer and antimicrobial assays and have found no activity, these data were added to the SI. Since we observed no activity, we have not discussed any further in terms of function/biological activity (we had already discussed potential functions). The co-author Maria Lúcia Sousa was included due to the new experimental work on bioassays.

4) What is the point of their phylogenetic analysis? Does it just show the widespread occurrence of BrTB

homologues? The authors may, ideally, show the function of one or two of these homologues. I would also suggest that the author move this section to the end of the manuscript.

We had carried out a phylogenetic analysis in order to illustrate the diversity and relationships between BrtB homologues and highlight opportunities for discovery. We understand that this is not critical to the main manuscript, also from reviewer #3's comments. Therefore, as suggested by this reviewer and reviewer #3, we have substantially shortened this section, moved it to the end of the manuscript and the figure to the SI and now show the function of the homologues that have been characterized in the tree.

Reviewer #2 (Remarks to the Author):

The manuscript "BrtB is a novel O-alkylating enzyme that generates fatty acid-bartoloside esters" describes a very interesting new enzyme activity. An enzyme that catalyses the Fischer (not Fisher) Speier esterification. The analytics of the enzyme catalysed reactions are thorough and convincing but the data on the enzyme are missing. Essentially one only learns that it has been cloned and the gene does express. The authors need to provide the biochemical characterisation of the enzyme too. Without this the study is incomplete.

We appreciate the fact that this reviewer finds the activity of BrtB interesting. In light of the reviewer's comments and also those from the other two reviewers, we have now provided more biochemical data for BrtB. The new version of the manuscript includes:

- a) selectivity data for the first and second esterification steps;
- b) substrate scope experiments regarding other potential nucleophiles, as well as variations to the alkyl chloride structure;
- c) activity dependence on temperature;
- d) we show that the two oxygen atoms of the fatty acid are incorporated into each ester bond formed, proving that the carboxylate is a nucleophile in this reaction;
- e) we show that when the enzymatic reaction is carried out in D₂O excess, no deuterium is incorporated into the products, precluding a two-step elimination / addition.

Reviewer #3 (Remarks to the Author):

The paper describes the discovery of an unusual O-alkylating enzyme BrtB, suggested to function through nucleophilic attack of a fatty acid carboxylate on halogenated bartolosides to produce a wide range of new natural product esters. Overall this is a nice story/discovery. BrtB is a novel enzyme and there is good characterisation/evidence of the products formed in vivo and from in vitro reactions. They also show the enzyme is fairly promiscuous with respect to the fatty acid substrate. In light of this I believe the paper would be of interest to those in the biosynthesis and enzymology field. However, there are a number of issues that would need to be addressed before publication:

The introduction suggests that there is only one other example in nature where a carboxylate acts as a nucleophile for ester formation (Fig 1a). Actually there are a number O-methyltransferases where a carboxylate substrate acts as a nucleophile with SAM in a SN2 reaction to form methyl esters which is directly analogous to the proposed reactivity of BrtB.

<https://www.ncbi.nlm.nih.gov/pmc/articles/PMC167163/>

<http://www.jbc.org/content/287/44/37010.full.pdf>

<https://aac.asm.org/content/58/2/950>

We apologize for this oversight, we did not consider this type of reactivity (as we noted also to reviewer #1) and thank the reviewer for pointing us towards the relevant literature. We now include this reactivity as precedence for carboxylate nucleophiles in carboxyester formation, added it to Fig. 1 together with also another example from an eukaryotic CMLE. We refocus our introduction and discussion to emphasize that the novelty of this reaction lies on the nature of the nucleophile (non-activated fatty acid) and its electrophilic alkyl halide partner.

The section in the middle of the paper discussing bioinformatics analysis and potential BGCs containing BrtB like enzymes is too long and does not come up with any useful hypothesis regarding what other similar esters might be out there and what their role might be in nature.

Our idea was to illustrate the potential for discovery of new natural products and eventually new enzymatic reactions, but we agree that this might be too descriptive and therefore better suited to the SI, as also noted by reviewer #1. We have moved this section into the SI and only briefly mention it in the final part of the main text.

They assume the reaction proceeds by an SN2 mechanism similar to haloalkane dehalogenases but provide no evidence to support this. They should determine the absolute configuration of the substrates and products, so that they can establish if the reaction proceeds with overall retention or inversion of stereochemistry. Inversion would support the idea of direct attack of the fatty acid carboxylate onto the alkylhalide and retention could indicate possible covalent-catalysis proceeding via initial attack of active site nucleophile (e.g. Asp or Glu as in haloalkane dehalogenases).

We thank the referee for this comment, it is true that we do not provide evidence towards SN2 and there are alternative mechanisms which we didn't rule out. We now do not mention SN2 as the likely mechanism and do not present any assumption that this is the case (as we did before, regarding haloalkane dehalogenases). We've also made some additional experiments that provides some, albeit limited, insight into the reaction, although we were unable to determine whether inversion of configuration occurs or not (as explained below).

The referee's comment led us to revisit the paper by Schultz (2019, Angew Chem) concerning CylK (BrtB homolog) and in which the authors state that it is unclear whether a SN1 or SN2 reaction is taking place. In that case, both substrates and products are clearly defined stereochemically. Based on this as well as on the reviewer's comments, we feel that it would be better not to speculate on the likelihood of a SN2-like reaction.

We understood that the reviewer meant that inversion of stereochemistry would support SN2 (but not exclude SN1), while retention would support a covalent intermediate with a BrtB amino acid side chain, similarly to what happens in haloalkane dehalogenases. Initially, we thought that the reviewer meant that an alcohol would be formed (as in haloalkane dehalogenases) from hydrolysis of the covalent intermediate, and this alcohol would then serve as the nucleophile in the esterification. But then this would not lead to retention of stereochemistry. Thus, we assumed that the reviewer meant that a covalent intermediate would be formed (by SN2) and then the

fatty acid carboxylate would be the nucleophile in another SN2 reaction with concomitant exit of the amino acid side chain from BrtB to generate the final ester with retention of configuration.

The challenge posed by devising and carrying out the non-trivial syntheses for the assignment of at least relative stereochemistry of one bartoloside and one ester thereof (ECD should not work, due to the lack of a nearby chromophore in the alkyl halides) led us to plan a degradative approach using bartoloside G and its monoester, with generation of aglycones and comparison of optical rotations, but we encountered problems during the stereocontrolled ester hydrolysis and the use of protecting groups and had to abandon that plan.

We envisioned that we could at least seek experimental proof for the hypothesis of the fatty acid carboxylate acting as a nucleophile. We could find (as had been shown in Fig. 3d in the previous manuscript) bartoloside A esterified with acetate and confirmed this in in vitro assays during the preparation of the revised manuscript, so we knew that acetate would be incorporated. Hence, we conducted in vitro assays with $^{18}\text{O}_2$ -labeled acetate and bartoloside A to show that two ^{18}O atoms are incorporated per ester bond formed, therefore proving that the fatty acid carboxylate is indeed a nucleophile in the reaction catalyzed by BrtB. These data were added to the manuscript, to Fig. 3b and to the SI.

One other mechanistic possibility was that the reaction mechanism involved an elimination reaction, followed by nucleophilic attack by the carboxylate onto the double bond. To test this hypothesis, we carried out a BrtB assay in D_2O -re-suspended buffer. We did not observe incorporation of deuterium into the products and therefore ruled out the possibility of such a mechanism. This is discussed in the manuscript and the data are shown in the SI.

We understand that we have fallen short of providing the requested mechanistic insight and, while we will certainly work towards this, we believe that it will require a substantial amount of additional work that is better suited for a future publication.

They use a lot of unsure language particularly in the conclusion – serendipitous, seems, suggests, likely, speculate, might, tempting. Whilst, I appreciate it is important to not be too speculative or overstate the case, the language does give the impression that the authors are not convinced by their conclusions.

We have tried to be more assertive throughout the text.

The paper suggests BrtB might be useful as an industrial biocatalyst, but does not provide any suggestions or examples of how such applications might be realised.

We now suggest that, due to its flexibility when it comes to fatty acid length, BrtB might be useful in transformations of bio-based resources such as oil and fat feedstocks.

The Figures are cluttered. For example, Fig 2 bottom right hand corner looks like it is linked to part d, but it is linked to c. Fig 2d is far too small (illegible) and provides little information. It should be removed with an expanded and more detailed version added to SI.

We have rearranged the figures to make them less cluttered. Figure 2D was removed and we kept the more legible phylogenetic tree that was already featured in the SI.

Reviewers' comments:

Reviewer #1 (Remarks to the Author):

The authors have carefully addressed my concerns and I do not have further comments on this manuscript.

Reviewer #2 (Remarks to the Author):

The manuscript "BrtB is a novel O-alkylating enzyme that generates fatty acid-bartoloside esters" has been greatly reworked and now describes a very interesting new enzyme activity in more detail.

The analytics of the enzyme catalysed reactions are thorough and have been improved and crucial labelling studies have been added that prove this to be the action of one enzyme. The data on the product characterisation are now even more convincing but the standard data on the enzyme are still missing. Essentially one still only learns that it has been cloned and the gene does express. The gels showing the purified enzyme do not show highly purified enzymes. But given the labelling studies it really seems to be the action of just one enzyme. The enzyme does not seem to be very active given the long reaction times but the authors should give time curves. 0, 6 and 24 h (page 22) just let the reader guess that the enzyme is not highly active. But with information such as V_{max} and K_m (should be possible with this enzyme preparation, k_{cat} not) a clearer picture can be obtained. Since no information about the conversion over time is given it is also not clear whether the enzyme is actually inactive at the end of the reaction. The study of pH and temperature optima might be part of a different study with pure enzyme that should then also include k_{cat} . And actually the authors might find out that the conditions they use for their enzyme assay are not the best for the enzyme.

Reviewer #3 (Remarks to the Author):

The authors have addressed the majority of the questions from the reviewers. I appreciate that it was difficult to establish the absolute configuration of substrates and products. However, the ^{18}O labelling experiments do provide some insight into the mechanism of the enzyme. Overall, the paper

has improved considerably in light of the modifications that have been made. I therefore recommend publication.

BrtB is a novel O-alkylating enzyme that generates fatty acid-bartoloside esters

Response to Reviewer comments (responses in blue color):

Reviewer #2 (Remarks to the Author):

The manuscript "BrtB is a novel O-alkylating enzyme that generates fatty acid-bartoloside esters" has been greatly reworked and now describes a very interesting new enzyme activity in more detail. The analytics of the enzyme catalysed reactions are thorough and have been improved and crucial labelling studies have been added that prove this to be the action of one enzyme. The data on the product characterisation are now even more convincing but the standard data on the enzyme are still missing. Essentially one still only learns that it has been cloned and the gene does express. The gels showing the purified enzyme do not show highly purified enzymes. But given the labelling studies it really seems to be the action of just one enzyme. The enzyme does not seem to be very active given the long reaction times but the authors should give time curves. 0, 6 and 24 h (page 22) just let the reader guess that the enzyme is not highly active. But with information such as V_{max} and K_m (should be possible with this enzyme preparation, k_{cat} not) a clearer picture can be obtained. Since no information about the conversion over time is given it is also not clear whether the enzyme is actually inactive at the end of the reaction. The study of pH and temperature optima might be part of a different study with pure enzyme that should then also include k_{cat} . And actually the authors might find out that the conditions they use for their enzyme assay are not the best for the enzyme.

We thank the reviewer for the suggestions to improve the characterization of the activity of BrtB. Accordingly, we have calculated K_m (apparent) and V_{max} (apparent) using varying concentrations of fatty acid (6-heptynoic acid, **2**). We used bartoloside G (**5**) as the other substrate, since bartoloside A (**1**) can be esterified in two positions. The reactions were analyzed by LC-HRESIMS and we used a standard curve of the monoester (**4**) to quantify the product formed in the assays. Our data showed a K_m , app of 234 μM and V_{max} , app of 0.13 $\mu\text{M}/\text{min}$ and we have added this information to the manuscript and the curve (presented below) was included in the SI.